# Factors affecting the biology of *Pachycrepoideus vindemmiae* (Hymenoptera: Pteromalidae), a parasitoid of spotted-wing drosophila (*Drosophila suzukii*)

**Cherre S. Bezerra Da Silva** ⬚*, **Briana E. Price, Alexander Soohoo-Hui, Vaughn M. Walton**

Department of Horticulture, Oregon State University, Corvallis, Oregon, United States of America

¤ Current address: Embrapa Algodão, Campina Grande, Paraíba, Brazil
* entomologista@gmail.com, cherre.silva@embrapa.br

**Data Availability Statement:** All relevant data are within the manuscript and its Supporting Information files.

## Abstract

*Pachycrepoideus vindemmiae* is a wasp that parasitizes and host-feeds on pupae of the invasive spotted-wing drosophila (SWD, *Drosophila suzukii*). Few studies have addressed interactions between these two species and little is known about the potential of this parasitoid as a biocontrol agent of SWD and the different variables that may affect it. Here, we investigated the impact of extrinsic and intrinsic factors on life-history traits of *P. vindemmiae*. Both constant (entire adulthood) and limited (30 minutes) supplies of water + honey, honey, or host increased parasitoid survival compared to controls (water or fasting). Water + honey caused the highest parasitoid survivals (35–60 days), independent of supply period, sex, and host availability. Females were intrinsically more resistant to water- and honey-deprivation than males, and host-feeding elevated such resistance even higher. Constant honey supply (either with or without water) supported the highest host-killing capacities (= capacity to kill hosts) (*ca*. 600 SWD pupae/wasp). However, in young females (4–9 days old), the impact of honey availability (with or without water) was insignificant while water deprivation (either with or without honey) caused the highest host-killing potential. This indicates that although sugar becomes a critical nutritional resource as females age, young females depend more on water than sugar to reproduce. Neither water nor honey affected the sex ratio of young females, but when we considered the entire adulthood, the availability of honey caused the lowest proportion of females (0.50), independent of water availability. Neither water nor honey affected parasitoid emergence rate (0.97), independent of female age. Based on survival and host-killing capacity, we conclude that *P. vindemmiae* has a tremendous biocontrol potential against SWD. Both limited and constant supply of water, sugar, and host increase parasitoid survival, while constant supply of water and/or honey enhance its host-killing potential and decrease sex ratio depending on maternal age.

**Funding:** This work was supported by the Oregon Blueberry Commission (VMW, CSBDS); Oregon Raspberry and Blackberry Commission (VMW, CSBDS); United States Department of Agriculture (USDA) – National Institute of Food and Agriculture (award #2015-51181-24252) (VMW); USDA – Organic Agriculture Research and Extension Initiative (#2014-51300-22238) (VMW); and Oregon State University Agricultural Research Foundation. The funders had no role in study design, data collection and analysis, decision to publish, or preparation of the manuscript.

**Competing interests:** The authors have declared that no competing interests exist.

## Introduction

Since 2008, spotted-wing drosophila (SWD), *Drosophila suzukii* (Diptera: Drosophilidae), has become a key pest of small fruits and cherries in the Americas and Europe [1–3]. The fly is native to southeast Asia and was first detected in North America (California) [4] and Europe (Italy and Spain) [5] in 2008, and in South America (Brazil) in 2013 [6]. Management relies greatly on broad-spectrum insecticides [7,8] and cultural tactics such as early/timely harvest and sanitation [9–11]. However, these approaches display limited efficacy, and are not environmentally or economically sustainable. For these reasons, there is a need for alternative control strategies that can be incorporated into a holistic integrated pest management (IPM) program for SWD [9,12]. Biological control provides such an alternative as it is relatively safe and compatible with other forms of pest control, constituting one of the main pillars of most IPM programs [13,14]. In the context of SWD, parasitic wasps are expected to play a key role in reducing field populations of this fly pest in newly invaded areas [3,9,12,15–17].

Besides causing reproductive effects that often culminate with host death (i.e., oviposition and host-feeding), parasitoids can negatively affect their hosts in ways that do not contribute to parasitoids' current or future reproduction. This is known as non-reproductive effects and they include host mutilation, pseudoparasitism, and aborted parasitism, among others [18]. Additionally, because of the low availability of food (e.g., hosts and/or sugars) and its spatial and temporal variability in the field, determining the effects of single feeding events in parasitoids is critical for a better understanding of their nutritional needs and hence the frequency at which they need to forage for food [19,20].

*Pachycrepoideus vindemmiae* Rondani (Hymenoptera: Pteromalidae) (Fig 1) is a solitary idiobiont synovigenic ectoparasitic wasp that attacks pupae of many fly (Diptera) species [21,22], including SWD [23–25]. Its use as a natural enemy of dipterous pests has been evaluated in various countries [26–28]. In Europe and the U.S., *P. vindemmiae* is amongst the only two parasitoid species that have been naturally found to successfully attack and kill SWD in the field [16]. Such natural occurrence suggests that this parasitoid is already adapted to the new host and has the potential to succeed in biological control programs of SWD in these invaded areas. Nevertheless, research exploring the interaction between *P. vindemmiae* and SWD is limited, and very little is known about the impact of intrinsic (e.g., sex, age) and extrinsic (e.g., nutritional resources) factors on such interaction.

Thorough review on the effects of carbohydrate supply on various life-history traits of hymenopteran parasitoids have been conducted [29]. Compared to sugars, the impact of water on the biology of parasitic wasps has received far less attention. Water is considered an essential nutrient for insects [30,31] and must be ingested frequently to compensate for dehydration [32,33]. In parasitic wasps, water is seldom considered to have any effect on survival or reproduction, and little is known about whether parasitoids forage for free water [34–40]. By practicing host-feeding a parasitoid can access nutrients for parasitoid maintenance, egg production, or both, including water and sugars [41,42]. In *P. vindemmiae*, host-feeding has been observed not only on pupae of SWD [43], but also on pupae of the housefly (*Musca domestica* L.) [26], and *Drosophila melanogaster* Meigen [44]. In SWD, host-feeding by *P. vindemmiae* can kill the host pupa [43]. When parasitizing pupae of the housefly, females of *P. vindemmiae* that were offered a constant supply of honey had greater longevity, offspring production, and sex ratio (i.e., proportion of female offspring) than females fed a water-only diet [36]. Honey extends longevity in *P. vindemmiae* depending on SWD pupae and water availability, but it has no effect on the reproduction of young females (4–8 days old). These females seek and benefit from ingesting free water, and they host-feed on SWD pupae as a water-intake strategy. Host-feeding extends their longevity, and increased host-feeding lead to greater offspring

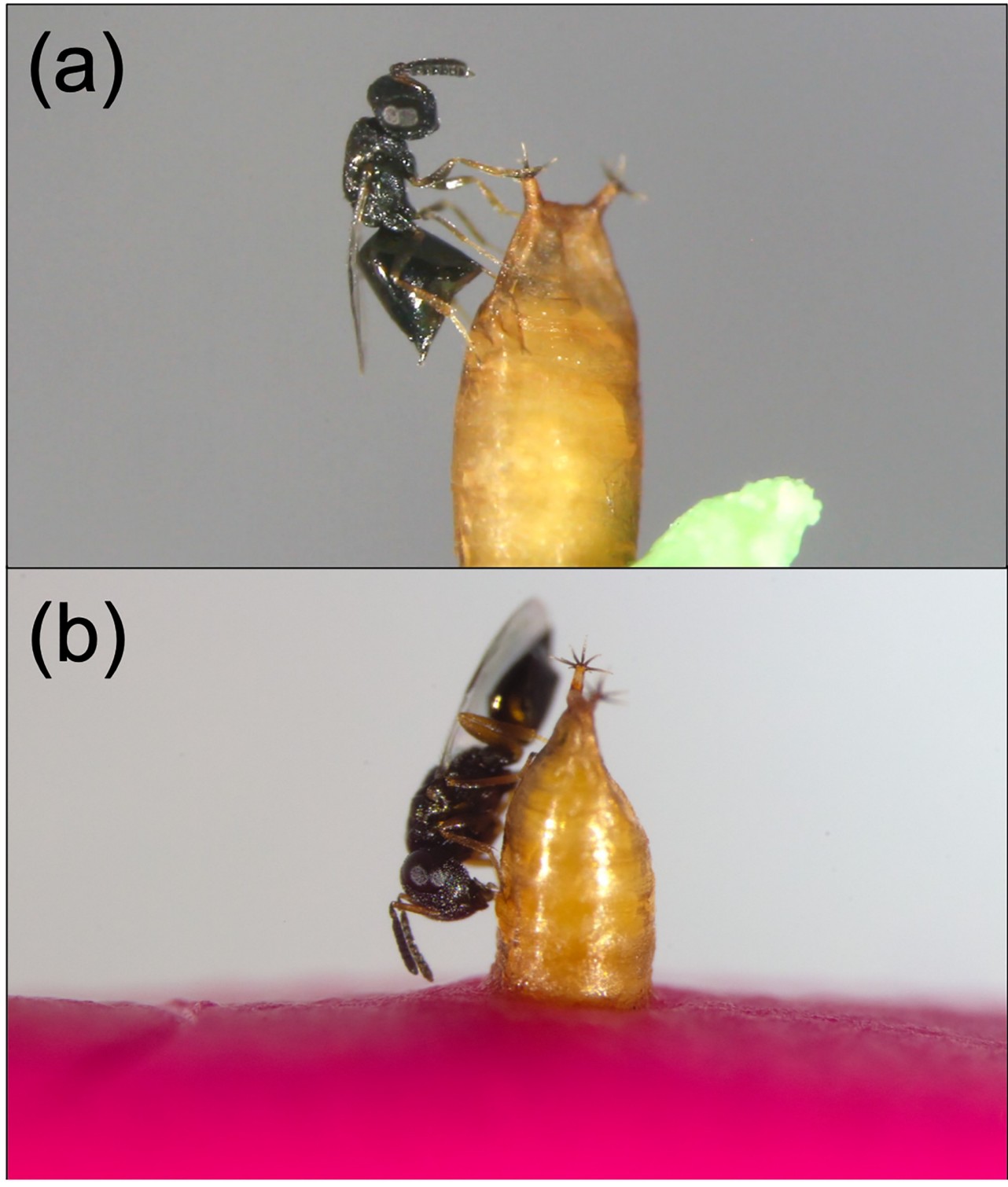

**Fig 1. *Pachycrepoideus vindemmiae* (Hymenoptera: Pteromalidae) attacking pupae of spotted-wing drosophila (SWD), *Drosophila suzukii* (Diptera: Drosophilidae), in the laboratory.** (a) Female wasp inserting her ovipositor through the SWD's pupal case and performing internal evaluation of the host prior oviposition. (b) Female wasp host-feeding on hemolymph of a SWD pupa following ovipositor withdrawal from the pupal case.

production [43]. Despite the studies on young females, comprehensive descriptions of the reproductive and non-reproductive effects of *P. vindemmiae* on SWD pupae throughout the parasitoid's entire life span are lacking. Measuring both effects is crucial for better understanding the full host-killing potential of the parasitoid against this invasive pest. Moreover, determining how nutrition affect such potential will indicate the parasitoid's water and food needs and may offer insights on habitat manipulation strategies.

In this study we investigated the impact of sex, age, and nutritional resources on the biology of *P. vindemmiae* reared on SWD. We hypothesized that water, honey, and host availability would affect the parasitoid's fecundity, emergence rate, sex ratio, host-feeding, and non-reproductive effects on SWD in a sex- and age-dependent manner. Additionally, we hypothesized that the impact of a limited (single event) and constant supply of those nutritional resources would increase the survival curves of adult parasitoids. We found that both a limited and constant availability of water, honey, and host increase the survival of *P. vindemmiae* compared to starved controls, and that females have tremendous survival and killing potential against SWD even in periods of water and sugar scarcity. By comparing data collected throughout the entire lifespan with that of early adulthood we show that, as opposed to water, sugar is not a critical resource for young *P. vindemmiae*, but its relevance to the parasitoid's host killing capacity and offspring sex ratio becomes evident as females age.

## Results

### Assay #1—Constant supply

**Survival curves.** After providing males and females of *P. vindemmiae* with a constant supply of different water and honey regimens, both in presence and absence of hosts, we found that the availability of water + honey resulted in significantly higher survival curves than honey alone, water alone, and fasting (no honey, no water), independent of sex and host availability (Males: P<0.0001, N = 8–9; Females: P<0.0001, N = 10–16; Females + host: P = 0.0049, N = 9–15) (Fig 2a and 2b, Table 1). In host absence, both female and male *P. vindemmiae* receiving honey displayed the second highest survival curve, followed by water and fasting, which did not significantly differ from each other. Females fed honey alone showed much greater survival than their male counterparts (P = 0.0027), but such difference became insignificant when the adults were offered water + honey (P = 0.2454) (Fig 2a and 2b, S1 Table). In host presence, no significant differences were found among survival of females offered the water, honey, and fasting treatments (Fig 2b, S1 Table). Host presence resulted in significantly lower survival curves than host absence in the honey (P = 0.0074) and water + honey (P = 0.0107) treatments. Conversely, host presence significantly increased survival compared to host absence in the water (P<0.0001) and fasting (P<0.0001) treatments (Fig 2b, S1 Table). The survival curves of host-provided females were either higher (water: P<0.0001, fasting: P<0.0001) or similar (honey, P = 0.1843) to those of males, with exception to water + honey where females had lower curves than males (P = 0.0019) (Fig 2a and 2b, S1 Table). As long as a honey source was available, the survival curves of host-deprived females were either higher (honey: P = 0.0027), or similar (water + honey: P = 0.2454) to those of males (Fig 2a and 2b, S1 Table). The average longevity of *P. vindemmiae* adults at each diet, sex, and host combination is shown in Table 2.

**Fecundity.** The parasitism rates of *P. vindemmiae* on SWD pupae were highest at 3–12 days old (the first 10 days of the study). Then, parasitism rates consistently declined independent of the diet. A steeper decline was however observed in the water and fasting diets compared to the other treatments (Fig 3a and 3b). The parasitism capacity of *P. vindemmiae* females was completely exhausted at ages *ca.* 26 (fasting), 31 (water), 35 (honey), and 40

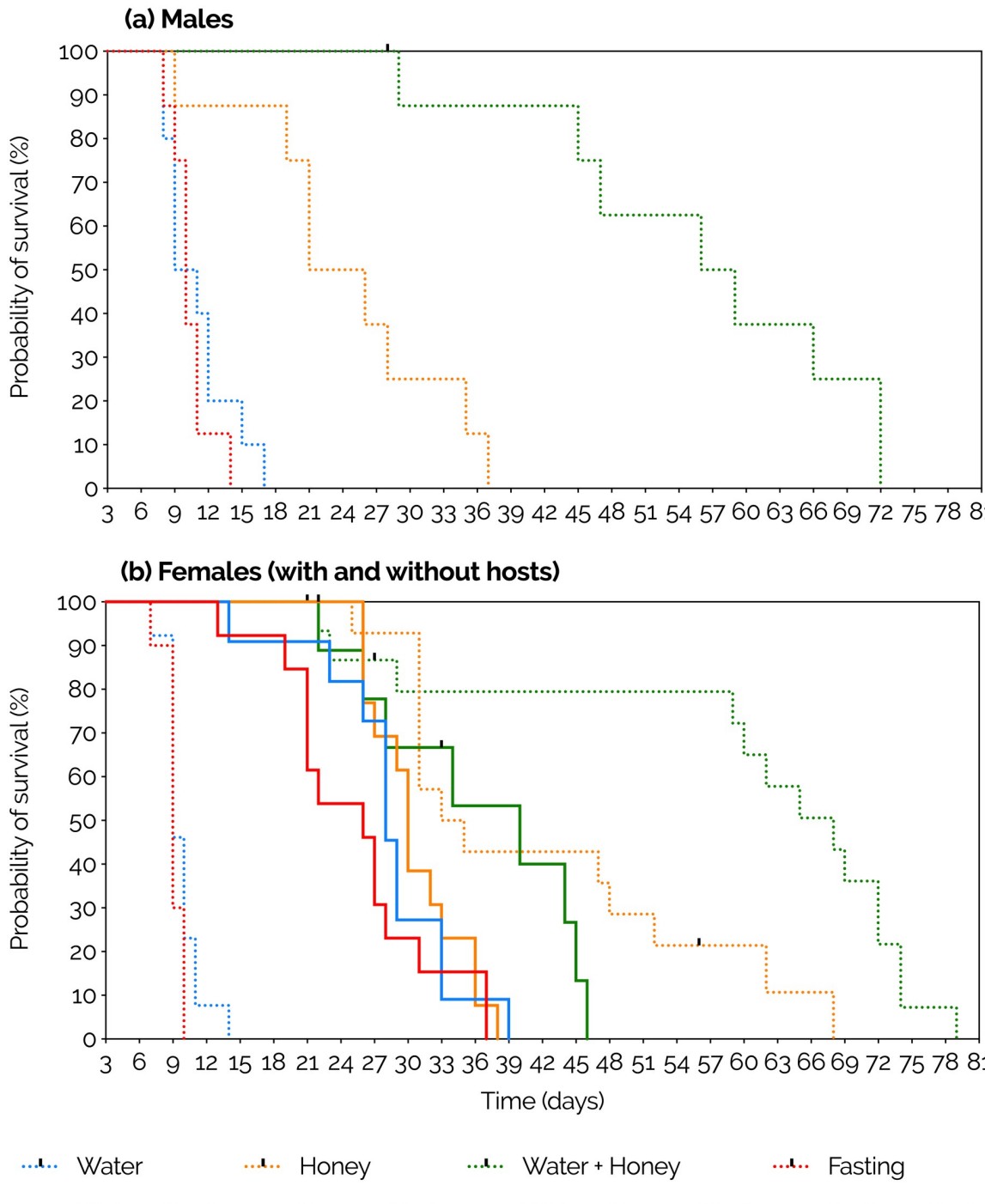

**Fig 2. Effects of constant supply of water, honey, and host on the survival curves of male (a) and female (b) adults of** *Pachycrepoideus vindemmiae* **(Hymenoptera: Pteromalidae) reared on pupae of spotted-wing drosophila (*Drosophila suzukii*) (Diptera: Drosophilidae).** Curves were estimated according to the Kaplan-Meier method. Dark dots represent censored data (either because wasp escaped or was accidentally killed when food, water, or hosts were being replaced). Curves were compared using the log-rank (Mantel-Cox) test (details in Table 1 and S1 Table). The numbers of replicates respectively for water, honey, water + honey, and fasting (no water, no honey) were 10, 8, 9, and 8 for males; 13, 14, 16, and 10 for females; and 11, 14, 10, and 12 for host-provided females. Each replicate was formed by a mated individual wasp.

**Table 1. Effects of diet (constant supply of water, honey, water + honey, or fasting) on survival curves of adult males, females, and host-provide females (females + host) of *Pachycrepoideus vindemmiae* (Hymenoptera: Pteromalidae) (24.1±0.4˚C, 62±8% R.H., and 14:10 L:D photoperiod).** Log-rank (Mantel-Cox) test (α = 0.05). Pupae of spotted-wing drosophila (*Drosophila suzukii*) (Diptera: Drosophilidae) were used as hosts. The numbers of replicates respectively for water, honey, water + honey, and fasting (no water, no honey) were 10, 8, 9, and 8 for males; 13, 14, 16, and 10 for females; and 11, 14, 10, and 12 for females + hosts. Each replicate was formed by a mated individual wasp.

| Group | $\chi^2$ | DF | P-value |
|---|---|---|---|
| Males | 39.78 | 3 | <0.0001**** |
| Females | 67.76 | 3 | <0.0001**** |
| Females + host | 12.87 | 3 | 0.0049*** |

***P<0.01

****P<0.0001

(water + honey). The likelihood of a wasp to be found alive at each of those ages was 48, 28, 40, and 40%, respectively, but it declined to zero *ca.* 3–11 days later (Fig 2b). When we considered the entire adulthood of *P. vindemmiae* females, honey provision strongly increased fecundity compared to honey deprivation, independent of water availability (P<0.0001, Table 3, Fig 4a). This scenario was completely reversed when we considered the early adulthood only (4–9 days old). Here, water deprivation increased fecundity relative to water provision, independent of honey availability (P = 0.0045, Table 3, Fig 5a).

**Miscellaneous attack.** Besides killing hosts directly through offspring production (Fig 1a, S1 Video), parasitoids also kill hosts through host-feeding (Fig 1b, S2 Video) and non-reproductive effects (pseudoparasitism, aborted parasitism, and mutilation [18]). Separating the latter two mortality factors when a SWD pupa is dissected can be very tricky and in this study we were unable to do so accurately. Hence, here we are considering host-feeding and non-reproductive effects as a single mortality factor, hereafter called "miscellaneous attack". The number of SWD pupae killed by *P. vindemmiae* through miscellaneous attack increased as wasps aged independent of diet (Fig 3c and 3d). When we considered the entire adulthood of *P. vindemmiae* females, miscellaneous attack was significantly increased in wasps fed honey compared to the honey deprived ones, independent of water availability (P = 0.0390, Table 3, Fig 4b). On the other hand, during *P. vindemmiae*'s early adulthood (4–9 days old), water-deprived wasps killed significantly more SWD pupae than their water-fed counterparts, independently of honey availability (P = 0.0211, Table 3, Fig 5b).

**Emergence rate.** Except for a few days along its entire adulthood, the emergence rate of *P. vindemmiae* on SWD pupae fluctuated around 100% independent of diet (Fig 3e). Neither water nor honey significantly affected this life-history trait when we considered the parasitoid's

**Table 2. Longevity (mean±SEM) of adult males, females, and host-provided females (females + hosts) of the pupal parasitoid, *Pachycrepoideus vindemmiae* (Hymenoptera: Pteromalidae), offered constant supply of water, honey, water + honey, or fasting (no honey, no water) (24.1±0.4˚C, 62±8% R.H., and 14:10 L:D photoperiod).** Pupae of spotted-wing drosophila (*Drosophila suzukii*) (Diptera: Drosophilidae) were used as hosts. The numbers of replicates respectively for water, honey, water + honey, and fasting were 10, 8, 9, and 8 for males; 13, 14, 16, and 10 for females; and 11, 14, 10, and 12 for females + hosts. Each replicate was formed by one mated individual wasp.

| Diet | Males | Females | Females + hosts |
|---|---|---|---|
| Water | 11.0±0.9 | 09.8±0.2 | 28.2±1.8 |
| Honey | 24.5±3.1 | 40.4±3.4 | 30.7±0.9 |
| Water + honey | 55.8±5.3 | 59.1±5.2 | 35.6±2.0 |
| Fasting | 10.4±0.5 | 09.1±0.3 | 25.4±1.6 |

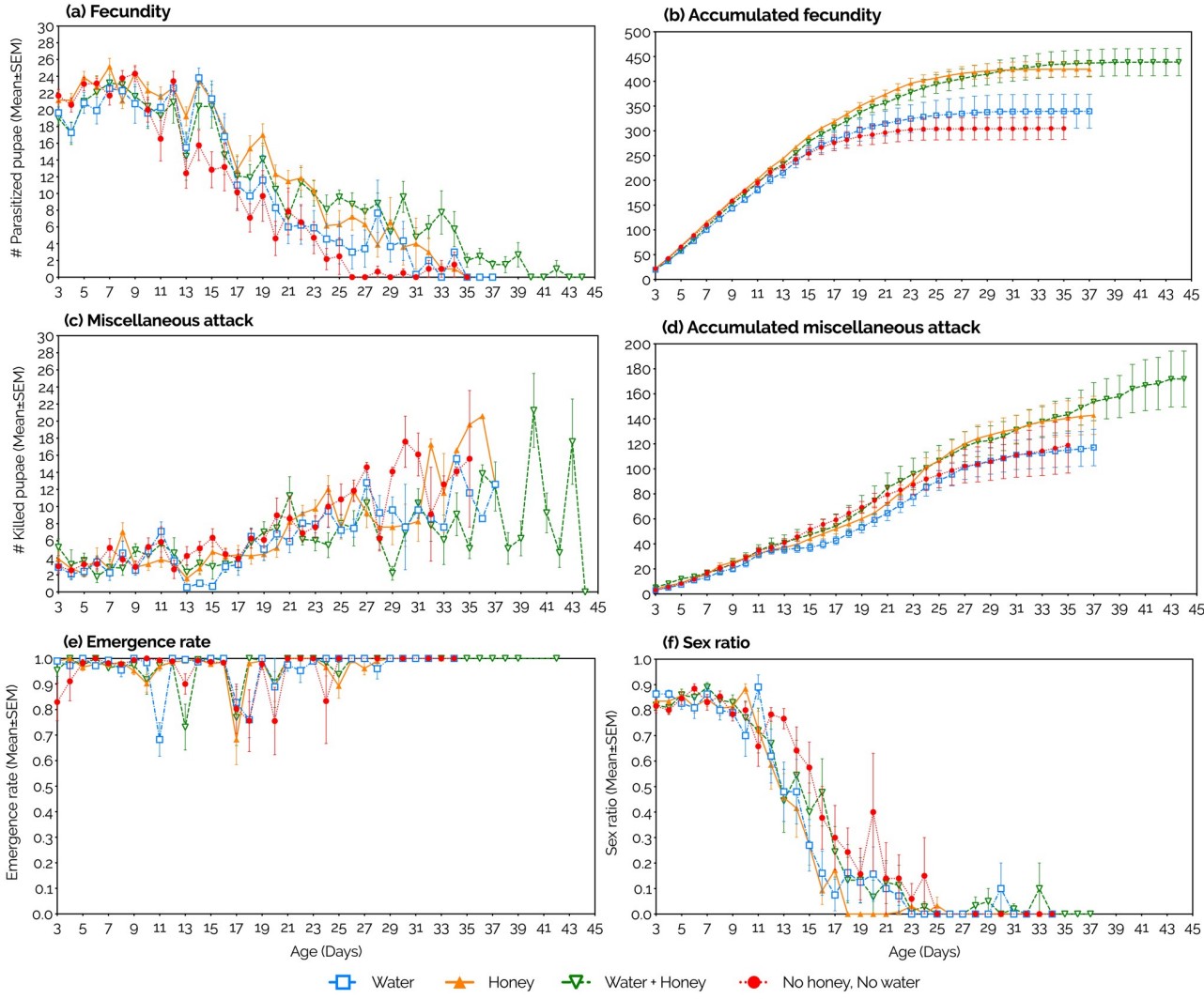

**Fig 3. Effects of constant supply of water and honey on life-history traits of females of *Pachycrepoideus vindemmiae* (Hymenoptera: Pteromalidae) reared on pupae of spotted-wing drosophila (SWD, *Drosophila suzukii*) (Diptera: Drosophilidae).** Traits examined include fecundity (a), cumulative fecundity (b), miscellaneous attack (c), cumulative miscellaneous attack (d), parasitoid emergence rate (e) and sex ratio (f). N = 11 (water), (14) honey, (10) water + honey, and (12) fasting. Each replicate was formed by one mated individual wasp.

entire lifespan (Honey: P = 0.2613, Water: P = 0.4589, Table 3, Fig 4c), or just the early adulthood (4–9 days old) (Honey: P = 0.7755, Water: P = 1835, Table 3, Fig 5c).

**Sex ratio.** The sex ratio (proportion of female offspring) of *P. vindemmiae* was highest in young wasps, whose offspring was composed of 80–90% female individuals through the first seven days of the experiment (wasps aged 3–9 days) in each of the four treatments. From day eight on, the proportion of females consistently declined independent of diet, reaching zero between the ages of 18 and 25 days (Fig 3f). When we considered the entire adulthood of *P. vindemmiae* females, sex ratio was strongly decreased in honey-fed wasps relative to their honey-deprived counterparts, independent of water availability (P<0.0001, Table 3, Fig 4d). When we considered early adulthood only (4–9 days old), neither water (P = 0.3677) nor honey (P = 0.3896) affected sex ratio (Table 3, Fig 5d).

**Table 3. Parametric Two-Way ANOVA to evaluate the effects of constant supply of water and honey on several life-history traits of *Pachycrepoideus vindemmiae* (Hymenoptera: Pteromalidae) reared on pupae of spotted-wing drosophila (*Drosophila suzukii*), throughout the parasitoid's early and entire adulthood.** N = 11 (water), (14) honey, (10) water + honey, and (12) fasting. Each replicate was formed by one mated individual wasp.

| Life-history trait | Age range | Source of variation | DFn, DFd | F | P-value |
|---|---|---|---|---|---|
| Fecundity | Early adulthood (4–9 days old) | Water | 1,44 | 8.969 | 0.0045** |
| | | Honey | 1,44 | 1.205 | 0.2784 NS |
| | | Interaction | 1,44 | 0.02397 | 0.8777 NS |
| | Entire adulthood | Water | 1,41 | 1.068 | 0.3076 NS |
| | | Honey | 1,41 | 19.22 | <0.0001**** |
| | | Interaction | 1,41 | 0.2097 | 0.6494 NS |
| Miscellaneous attacks | Early adulthood (4–9 days old) | Water | 1,41 | 5.748 | 0.0211* |
| | | Honey | 1,41 | 0.0007962 | 0.9776 NS |
| | | Interaction | 1,41 | 0.4119 | 0.5246 NS |
| | Entire adulthood | Water | 1,41 | 0.2126 | 0.6472 NS |
| | | Honey | 1,41 | 4.547 | 0.0390* |
| | | Interaction | 1,41 | 0.3465 | 0.5593 NS |
| Emergence rate | Early adulthood (4–9 days old) | Water | 1,40 | 0.1835 | 0.6707 NS |
| | | Honey | 1,40 | 0.7755 | 0.3838 NS |
| | | Interaction | 1,40 | 0.1153 | 0.7360 NS |
| | Entire adulthood | Water | 1,44 | 0.5582 | 0.4589 NS |
| | | Honey | 1,44 | 1.295 | 0.2613 NS |
| | | Interaction | 1,44 | 3.218 | 0.0797 NS |
| Sex ratio | Early adulthood (4–9 days old) | Water | 1, 44 | 0.8284 | 0.3677 NS |
| | | Honey | 1,44 | 0.755 | 0.3896 NS |
| | | Interaction | 1,44 | 0.563 | 0.4571 NS |
| | Entire adulthood | Water | 1,41 | 1.558 | 0.2191 NS |
| | | Honey | 1,41 | 19.01 | <0.0001**** |
| | | Interaction | 1,41 | 3.724 | 0.0606 NS |

*P≤0.05

**P≤0.01

****P<0.0001

NS = non-significant effect.

## Assay #2—Limited supply

**Survival curves.** Diet had a significant effect on adult survival ($\chi^2$ = 166.4, DF = 4, P<0.0001). Water + honey caused the highest survival curve (P<0.0001), while water alone and fasting caused the lowest (P<0.0001). No significant differences were found neither between honey and host (P = 0.4821) nor between water and fasting (P = 0.1148) (Fig 6, S1 Table). The longevity (mean±SEM) of *P. vindemmiae* females was 6.6±0.11 for water (N = 94), 7.8±0.27 for honey (N = 47), 8.7±0.16 for water + honey (N = 95), 6.4±0.16 for fasting (no water, no honey) (N = 39), and 7.9±0.18 for host (SWD pupa) (N = 70). Each replicate was formed by one mated individual wasp.

## Discussion

Sex can determine a parasitoid's lifespan [45], especially when males and females differ in the way they allocate nutritional resources to survival and reproduction [46]. In our study, the impact of water and honey supply on adult *P. vindemmiae* was similar (but yet different) between males and females. For example, when observed separately, both sexes were highly

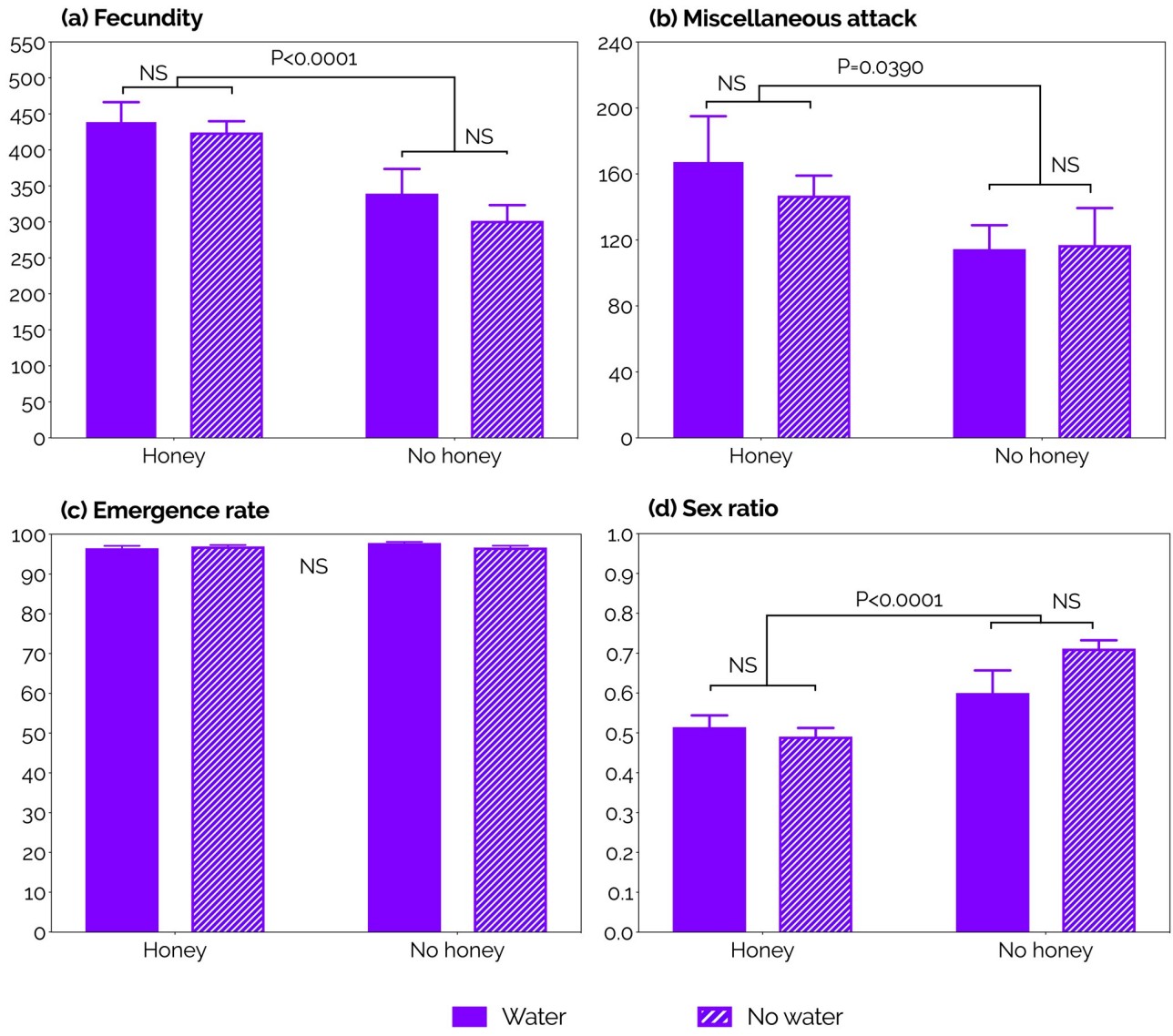

**Fig 4. Effects of constant supply of water and honey on the (a) fecundity, (b) miscellaneous attack, (c) emergence rate, and (d) sex ratio of** *Pachycrepoideus vindemmiae* **(Hymenoptera: Pteromalidae) reared on pupae of spotted-wing drosophila (SWD,** *Drosophila suzukii*) **(Diptera: Drosophilidae) for the parasitoid's entire adulthood.** P-values were calculated by Two-Way ANOVA; NS = no significant effect. N = 11 (water), (14) honey, (10) water + honey, and (12) fasting. Each replicate was formed by one mated individual wasp.

resilient in terms of survival. Offered water + honey they lived for up to 72–79 days, and even those individuals that were kept in complete starvation lived for 6–11 days. This lifespan is long relative to other parasitoid species and to a Chinese population of *P. vindemmiae* which under starvation only lived for 2.7 days [35,36,38,40]. Water + honey caused the highest survival curves, while water alone and fasting produced the lowest ones, independent of sex. However, by comparing survival curves between males and females we found that overall females live either same or longer than males, especially when water or sugars are scarce. First, honey-fed, host-deprived females survived better than males when water was scarce, but such a difference was diluted when water was offered, meaning that males are inherently more sensitive to water deprivation than females. Second, honey-deprived, host-provided females survived much better than males, but such advantage was suppressed when honey was supplied, i.e.,

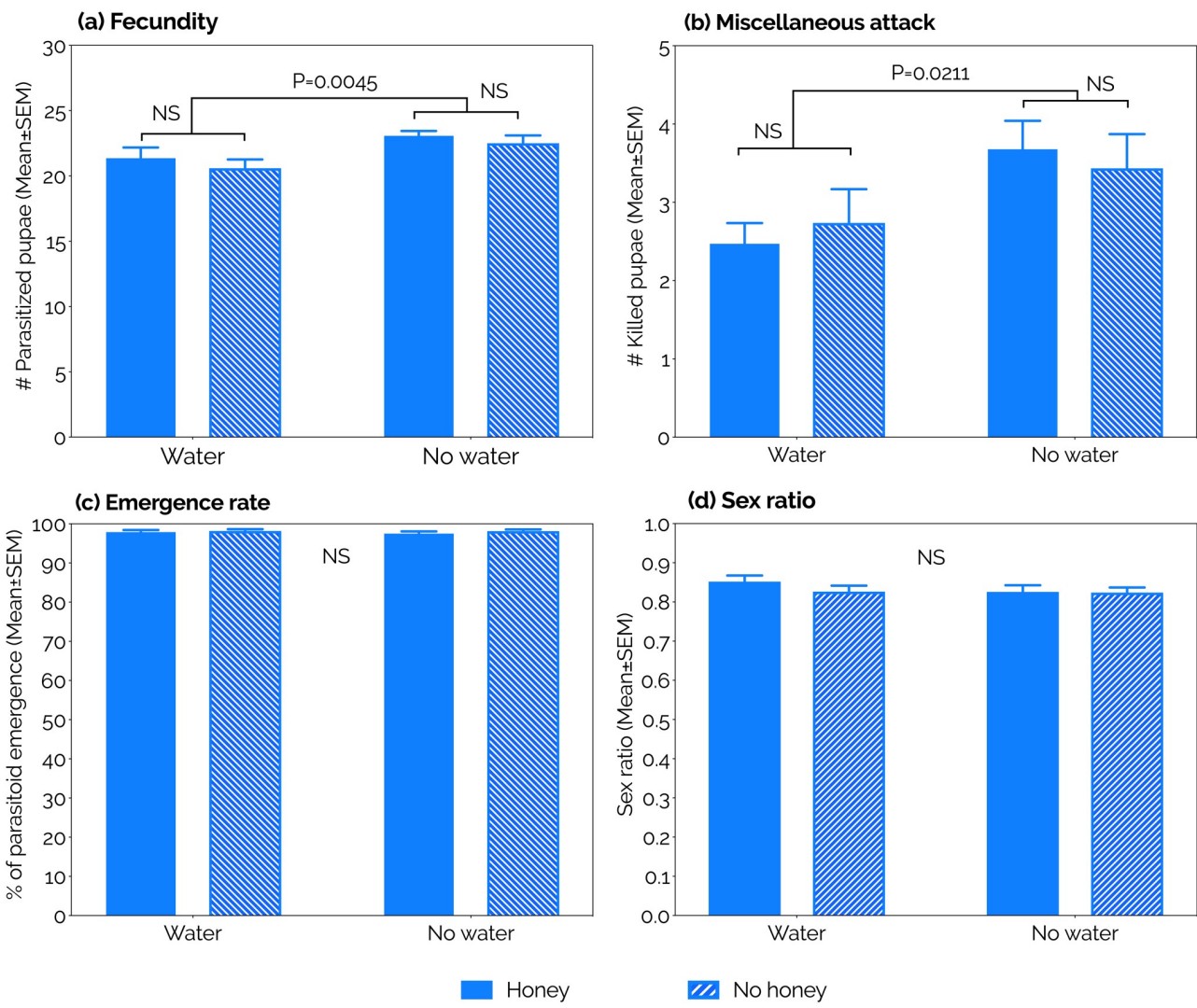

**Fig 5. Effects of constant supply of water and honey on the (a) fecundity, (b) miscellaneous attack, (c) emergence rate, and (d) sex ratio of** *Pachycrepoideus vindemmiae* **(Hymenoptera: Pteromalidae) reared on pupae of spotted-wing drosophila (SWD,** *Drosophila suzukii***) (Diptera: Drosophilidae) for the parasitoid's early adulthood (4–9 days old).** P-values were calculated by Two-Way ANOVA; NS = no significant effect. N = 11 (water), (14) honey, (10) water + honey, and (12) fasting. Each replicate was formed by one mated individual wasp.

males are more sensitive and hence more dependent on alternative sources of sugars than females. Taken together, these findings indicate that females are more likely to survive periods of water and sugar scarcity in the field than males, not only because females were intrinsically less sensitive to shortage of those resources, but also because females can supplement their nutritional needs with host-feeding.

It was clear that *P. vindemmiae*'s survival was highly influenced by both limited and constant supply of water and honey, especially when hosts were not available. The impact of water, honey, and host availability on female survival was very consistent between the constant and limited supply experiments (assays #1 and #2, respectively). Such consistency attests that feeding on water, sugars, and hosts have both immediate and cumulative benefits on the adult parasitoids. A single bout of feeding on those resources was enough to extend female lifespan by 1.2–2.1 days relative to starved individuals, likely more than enough time for the wasps to find

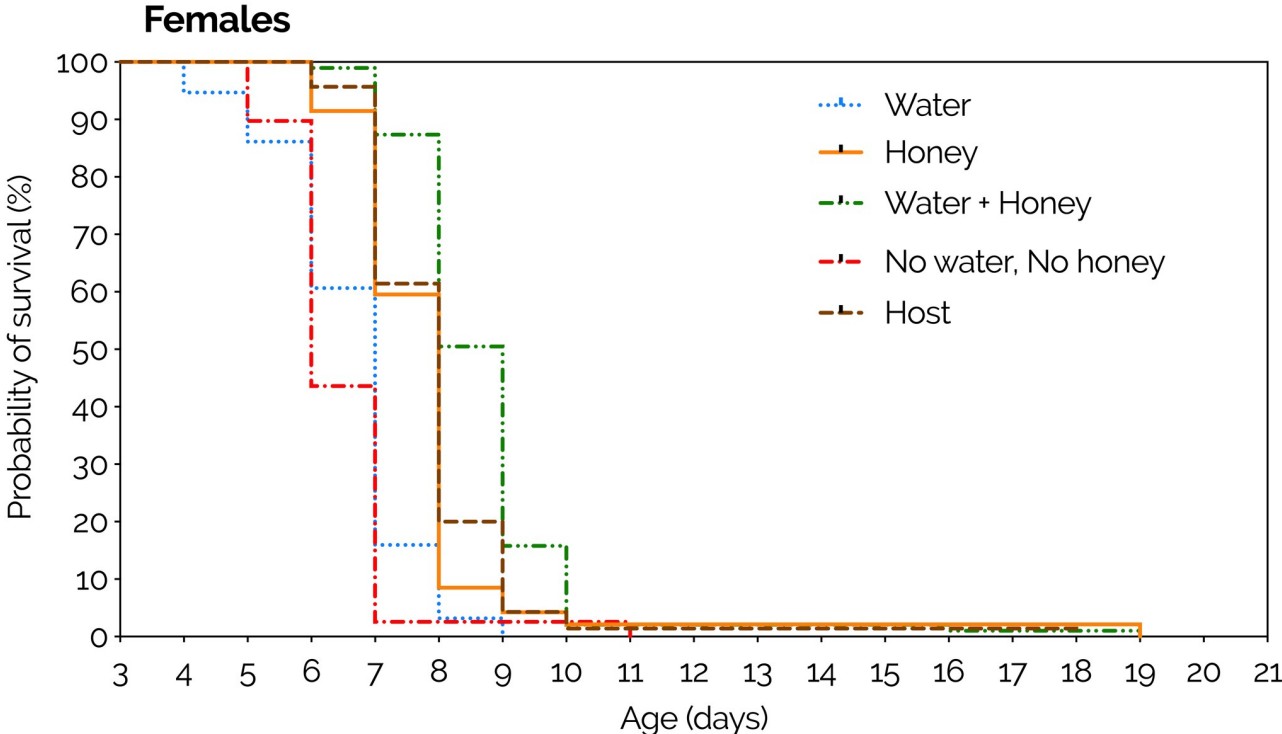

**Fig 6. Effects of limited supply (30 minutes) of water, honey, and host on survival curves of females of *Pachycrepoideus vindemmiae* (Hymenoptera: Pteromalidae) reared on pupae of spotted-wing drosophila (SWD, *Drosophila suzukii*) (Diptera: Drosophilidae).** Curves were estimated according to the Kaplan-Meier method. Curves were compared by the log-rank (Mantel-Cox) test (details in S1 Table). N = 94 (water), 47 (honey), 95 (water + honey), 39 (no water, no honey), and 70 (host, SWD pupa). Each replicate was formed by one mated individual wasp.

a next meal. Curiously, a small fraction of those females (0.5–1%) lived for 17–19 days following a single (30-minute) feeding bout on water + honey, honey, or host. This is 3x longer than the majority of their counterparts that were fed the same diets, highlighting important phenotypical plasticity regarding nutritional requirements in this parasitoid species, which may contribute to the reestablishment of *P. vindemmiae*'s populations in the field even after extended periods of food scarcity. These findings indicate that both starvation and dehydration strongly reduce parasitoid survival, hence *P. vindemmiae* must periodically search for sources of water and food in the field. Because foraging strategies are under strong selection pressure to minimize the risks to survival and maximize the nutrient gain from feeding [47], it is likely that the wasps will prefer sources where both water and sugars are combined in an ideal proportion to optimize foraging. Studies exploring the response of *P. vindemmiae* to different concentrations of sugars could offer insights on the parasitoid's sugar-water ratio preferences.

Our data indicates that female *P. vindemmiae* showed a tremendous killing potential against SWD. A single female wasp was capable of attacking between *ca.* 440 and *ca.* 600 SWD pupae throughout her lifespan, depending on diet. Such high killing potential was achieved through a combination of *ca.* 75% parasitism and *ca.* 25% miscellaneous attack (i.e., host-feeding, mutilation, pseudoparasitism, and aborted parasitism). These data were achieved under controlled and optimal conditions in laboratory experiments, but they nevertheless indicate the potential impacts of *P. vindemmiae* on SWD populations. A constant supply of honey led to the highest parasitism and miscellaneous attack rates independent of water availability, showing that sugar plays a key role in the total (i.e., entire adulthood) host-killing capacity of *P. vindemmiae*. Curiously, such sugar effect was not observed in young females (4–9 days old). At this age,

water (and not honey) was the critical factor affecting parasitism and miscellaneous attack as both rates were highest in water-deprived as opposed to water-fed individuals. We previously showed that water deprivation results in a significant increase in the host-killing capacity of young *P. vindemmiae* [43]. The current study, however, shows different host-killing results between early and entire adulthood, indicating that the water and sugar needs of adult *P. vindemmiae* females vary over their lifespan. In other words, our data in the current study demonstrates that females depend less on water and more on sugars for reproduction as they age. It should be noted, however, that an adequate supply of water + honey produced higher survival curves than any other diet, including honey alone. This finding suggests that even though the importance of water for parasitoid host-killing capacity decreases with the wasp's age, water can still be considered an important resource to parasitoid survival throughout its lifespan.

At least one third of all females exhausted their parasitism capacity many days before death, independent of water and sugar availability. Meanwhile, miscellaneous attack clearly rose, indicating that parasitism cessation was attributed to a lack of mature eggs rather than an inability to seek and drill hosts (i.e., exhaustion of maintenance energy). Fasted and water-fed females were the first ones to end parasitism, followed by wasps fed honey and water + honey, which is consistent not only with the survival curves, but also with the total (entire adulthood) fecundity and miscellaneous attack. The rise of miscellaneous attack as the wasps aged contributed to the maintenance of a consistent host-killing potential despite the parasitoid's declining fecundity. This finding seems plausible if we consider that (1) females survived on average 10–11 days without any food, (2) host-feeding plays a major role in miscellaneous attacks [43], and (3) the hemolymph of each SWD pupa extended female survival by 1.3 day. In other words, as females got older, they exhausted their nutritional reserves and thus increased host-feeding (and hence miscellaneous attack) as a means to compensate for the nutritional loss. Host-feeding increment as a function of parasitoid age has already been reported [48]. However, simply increasing host-feeding as a means to access nutritional resources and extend longevity does not explain such a high increment in miscellaneous attack. This is because, as previously explained, a single SWD pupa was enough to extend a female's lifespan by more than one day, while the increment in miscellaneous attack was elevated from *ca.* four pupae/day in females aged 15 days to *ca.* 12 pupae/day in females aged 30 days. Hence, nonreproductive effects such as mutilation, pseudoparasitism, and aborted parasitism, which can cause host death without contributing to current or future reproduction [18], must have played a major role in such increment. It is really interesting to observe that even though *P. vindemmiae*'s capacity of producing offspring declines or is even exhausted, the parasitoid still seeks out, exploit, and kill hosts thus contributing to biological control of SWD.

Determining where parasitoids invest the nutrients acquired from host-feeding contributes to answering one of the main questions regarding the reproductive success of parasitoids: the trade-off between current and future reproduction (i.e., to use a host to produce an offspring or to host-feed, respectively) [48–50]. It has been demonstrated that *P. vindemmiae* practice concurrent host-feeding (i.e., use the same host for both egg-laying and host-feeding) and that host-feeding by a newly-emerged female has no cost to her offspring neither in terms of survivorship nor size [44], suggesting that there is no trade-off between current and future reproduction for *P. vindemmiae*. Indeed, our results support such hypothesis since host-feeding by *P. vindemmiae* on SWD pupae was increased by water deprivation [43] while offspring emergence was not (Figs 4c and 5c), i.e., host-feeding has no effect on parasitoid offspring survivorship. However, advanced age and dehydration resulted in an increase in the rate and time invested in host-feeding, as well as the amount of consumed hemolymph [43,48], lowering the rate of emergence of parasitoid offspring (Bezerra Da Silva et al., unpublished data). Hence,

the trade-off between current and future two reproduction can be a constraint for *P. vindemmiae* depending on maternal age and nutritional status. Because host-feeding provides females of this species with nutrients for both reproduction and adult maintenance (Figs 2b, 3a, 3b, 4a, 5a and 6), then both the egg load and remaining life time of a female will dictate whether she uses a SWD pupa to produce an offspring or to host-feed.

When provided with water + honey, each parasitoid was able to parasitize 438 SWD pupae throughout a 36-day lifespan, with fecundity smoothly declining as wasps aged. These results very much differ from those reported by Rossi Stacconi et al. (2015), where each honey-water-fed female of *P. vindemmiae* parasitized only 78 SWD pupae throughout its lifetime of just 22 days, with a sharp decline in parasitism capacity just before death [51]. Even though the two studies maintained the wasps in very similar climatic conditions, the daily host supply regime differed. We offered 30 hosts per day, compared to 5 in Rossi Stacconi et al 2015. This methodological divergence likely resulted in the striking differences in total fecundity and pattern of parasitism exhaustion between the studies, but it does not explain the difference regarding lifespan. Because there is a trade-off between reproduction and survival, with investment in reproduction often reducing parasitoid survival [42,43,52], and the investment in this activity was 5.5x higher in our study than in Rossi Stacconi et al., it is surprising that the former reports a much greater survival than the latter.

The rate of emergence of *P. vindemmiae* was nearly 100%, independent of the diet and female age. This implies that pupae of SWD can be considered highly suitable as hosts for *P. vindemmiae*, especially relative to other fly species such as *Ceratitis capitata*, *Bactrocera latifrons*, *B. cucurbitae*, and *M. domestica*, which supported adult parasitoid emergence ranging from 35 to a maximum of 85% [21,26,53]. In our study, the high emergence rates throughout female lifespan independent of diet also suggest that neither the nutritional status nor the age of *P. vindemmiae* females affect their ability to lay viable eggs, as has been suggested for other parasitoid species [38,54,55]. Our results should be considered with parsimony however, because even though dead larvae and pupae of *P. vindemmiae* were easily spotted upon dissection, the same was not true for eggs due to their small size, translucid color, and dehydrated state. Hence, there is a chance that at least part of the high emergence rates observed in our study is an overestimation. As opposed to emergence rate, the impact of female age on offspring sex ratio was very clear, because the proportion of female offspring consistently declined starting in mothers aged *ca.* 10 days, independent of diet. Nevertheless, such decline was even stronger in honey-fed wasps in comparison to their honey-deprived counterparts, showing that sugar availability to mothers did have an effect on offspring sex ratio. Both maternal age and diet are known to affect the offspring sex ratios in parasitic wasps [56]. But because in our study the effect of honey on offspring sex ratio was observed exclusively when we considered the entire adulthood (i.e., not observed in young females), and also because honey-fed wasps were more fecund than wasps deprived of honey, it is clear that the effect of sugar on offspring sex ratio is related to maternal age instead of their diets. These females mated during the first 2 days following emergence but were then secluded from day 3. Under these conditions, females lacked an opportunity to re-mate and thus replenish their sperm storage as egg laying took place. By being more fecund (i.e., laying more eggs), water + honey- and honey-fed females had greater opportunities to deplete sperm storage, consequently producing proportionally fewer females than sugar-deprived parasitoids (water and fasting). These findings indicate that females of *P. vindemmiae* need to re-mate multiple times throughout their adulthood if they are to produce female offspring, independent of their nutritional status, especially if they are highly prolific. In laboratory colonies and mass rearing, this means that fertile males should be present and mate with females throughout their lifetime if the foundresses are going to be kept for longer than 10 days following emergence. In the field, it is likely that

prolific females will invest time and energy in re-mating as they age and their sperm storage is depleted.

## Conclusions

This study demonstrates that *P. vindemmiae* is long-lived and can significantly contribute towards SWD biocontrol through a combination of parasitism and miscellaneous attack (host-feeding, mutilation, pseudoparasitism, and aborted parasitism). We showed that limited and constant supplies of water and honey affect the survival of male and female wasps, as well as their parasitism capacity, sex ratio, and miscellaneous attack, with clear consequences for SWD mortality. Males are more sensitive to water and sugar scarcity than females. In the absence of water and sugars, females of *P. vindemmiae* can host-feed and rely exclusively on pupae of SWD to extensively extend their survival and increase their parasitism capacity. Even a single bout of feeding on hosts, sugar, and especially on both water and sugar, significantly extended the wasp's survival. Constant supplies of water and sugars in an environment with no hosts can result in survival of up to 72 days, allowing maintenance of parasitoid populations and resumed parasitism when hosts once again become available. But even if none of those resources are available, both males and females can survive for many days under complete starvation. Taken together, these characteristics demonstrate the high resilience, adaptibility and biocontrol potential of *P. vindemmiae* against SWD, proving their increased likelihood of surviving unfavorable periods of water, food, and host scarcity in both the laboratory and field. These findings open many opportunities for additional research. These include additional comparative studies with imported specialist parasitoids of SWD [57,58], controlled field studies to determine how nutrient supply can result in improved biocontrol [59,60], and studies to determine impacts of horticultural practices such as irrigation on the parasitoid efficacy [17].

## Material and methods

### Insects

Colonies of *D. suzukii* and *P. vindemmiae* have been maintained in a climate-controlled chamber in the laboratory since 2009 (24.1±0.4°C, 62±8% R.H., and 14:10 L:D photoperiod). The SWD colonies were started from insects kindly provided by the USDA-ARS (Corvallis, OR, USA). SWD flies from the Willamette Valley and Columbia River Gorge of Oregon have been introduced periodically to maintain genetic diversity. SWD larvae were fed a cornmeal diet while adults were offered *ad libitum* sucrose solution (10% w/w) and brewer's yeast [61]. Our *P. vindemmiae* colonies started in 2013 when adult wasps emerged from SWD pupae collected in sentinel traps in the Willamette Valley and Columbia River Gorge of Oregon [24]. Adult parasitoids were kept in Bugdorm mesh cages (32.5x32.5x32.5 cm) (Bioquip Products, Rancho Dominguez, CA, USA), fed streaks of honey on a Petri dish, and offered a pint container with a sponge soaked in de-ionized water. Once a week, *ca*. 1,500 fresh *D. suzukii* pupae (1–2 days old) were rinsed with tap water and left to dry for 1h at room temperature in a fume hood. The pupae were sprinkled on cardboard pieces (7x14 cm) freshly coated with paper glue. The pupae-carrying cardboard pieces were then left do dry for 1h in a fume hood at room temperature. Subsequently, they were inserted in the Bugdorm cages containing adult *P. vindemmiae* and exposed to parasitism for 24 h. The cardboard pieces were vigorously shaken to disperse adult wasps and immediately withdrawn from the cages. Adult wasps that eventually remained on the cardboard/pupae were aspirated and released back into the parasitoid cages. Each cardboard piece carrying parasitized SWD pupae was cut in two halves (7 x 7 cm each) and deposited in a pint container covered with a screen lid. The containers were maintained in the

climate-controlled chamber for about 17–20 days until parasitoid emergence. Male and female adults were released together into new Bugdorm cages to restart the cycle.

### Assay #1—Constant supply

Newly emerged (<24h) males and females of *P. vindemmiae* were allowed to mate in a Bug-dorm cage for 2 days. From emergence through mating period individuals did not have access to sugar, water or hosts. On day 3, wasps were individualized in flat-bottomed tubes (2x9 cm) (Genesee Scientific, San Diego, CA, USA) and offered (1) water, (2) honey, (3) water + honey, (4) no water, no honey (fasting), (5) water + hosts, (6) honey + hosts, (7) water + honey + hosts, and (8) no water, no honey + hosts, for their entire lives. Honey was offered as a small drop placed onto the tube wall. Water was provided from a thin (3mm wide) piece of filter paper extending from a hole made at the tip of a 2-mL Eppendorf tube filled with de-ionized water [43]. Treatments 1–4 (i.e., host absence) were tested on both males and females. Treatments 5–8 (i.e., host presence) were tested on females only since males of *P. vindemmiae* do not exploit hosts. Each host-provided female received 30 fresh (<2 days old) SWD pupae/day. Before exposure to females, the pupae were gently rinsed in de-ionized water to remove corn-meal and larval excrement residues. Using a fine camel brush and de-ionized water, the 30 pupae were carefully attached to a piece of paper towel (2 x 2 cm) and set aside to dry for 1h at room temperature. Only then the pupae-carrying paper pieces were offered to the wasps. The numbers of replicates respectively for water, honey, water + honey, and fasting were 10, 8, 9, and 8 for males; 13, 14, 16, and 10 for females in host absence; and 11, 14, 10, and 12 for females in host presence. Each replicate was formed by one mated individual wasp. Daily, 5–8 paper towel pieces (N = 262 pieces for the entire experiment), each carrying 30 SWD pupae, were randomly selected to serve as an unexposed control, i.e., to never be offered to *P. vindem-miae* and thus allow assessment of the natural SWD mortality. These pupae were held under the same conditions as the parasitoid-exposed pupae. To control for potential differences in the relative humidity among treatments that were water provided (1, 3, 5, and 7) and water deprived (2, 4, 6, and 8), the same water source was provided within all tubes independent of treatment. However, for the water-deprived treatments, the tip of the Eppendorf tube was covered with a screen cap to prevent wasps from reaching the moistened paper while still allowing vaporized moisture to dissipate within the tube [43].

Each day, the 30 parasitoid-exposed SWD pupae from each tube (treatments 5–8) were replaced by 30 fresh pupae and parasitoid mortality was recorded. Water and honey were resupplied as needed. The parasitoid exposed pupae were transferred to a new tube and kept in the climate-controlled chamber until adult emergence (24.1±0.4°C, 62±8% R.H., and 14:10 L:D photoperiod). Emerged flies and parasitoids were counted under the stereomicroscope (Leica S8AP0, Leica Microsystems Inc., Buffalo Grove, IL, USA). All SWD pupal cases that did not show emergence holes were dissected to determine whether parasitism did occur. This was done by looking for dead larva or pupa of *P. vindemmiae* inside each SWD pupal case. Parasit-oid fecundity was calculated as the total number of SWD pupal cases containing either a dead immature parasitoid or a wasp emergence hole. Parasitoid emergence rate was the number of emerged wasps/fecundity. Sex ratio was the number of female wasps/total wasps. Additionally, we calculated the number of SWD pupae killed by a combination of host-feeding and non-reproductive effects, since these are two important mortality factors but discerning one from the other with accuracy is often not possible [18]. The combination of host-feeding and non-reproductive effects in this study is referred to as "miscellaneous attack". It was calculated as the number of SWD pupal cases showing neither an emergence hole nor a dead immature par-asitoid minus natural SWD mortality.

Because young female parasitoids may have specific nutritional needs [48], all the above mentioned parameters were measured considering both entire and early adulthood (4–9 days old). One- and two-days old wasps were not used in the experiment because those ages were reserved for mating, as previously explained. Three-day olds were excluded because at that age females still carry the egg load developed with resources acquired at the larval stage [44].

## Assay #2—Limited supply

Two-days-old mated females of *P. vindemmiae* were split in five groups and offered (1) water, (2) honey, (3) water + honey, (4) no water no honey (fasting), or (5) host (N = 94, 47, 95, 39, and 70, respectively) for 30 minutes. Only females that were naïve to sugar, water, and hosts were used. In the first four groups, honey and water were offered in flat tubes as described in *Assay 1*. In the fifth group, each female wasp was individualized in a clear gelatinous capsule (#00) and offered a single host (<2-days-old SWD pupa). From a previous study [43], 30 minutes from the moment when each wasp first touched the host, allowed sufficient time for *P. vindemmiae* to drill a hole in a SWD's pupal case to host-feed and leave the host. Similarly, preliminary observations showed that 30 minutes was long enough for adult *P. vindemmiae* to exploit and leave the water and honey sources. After feeding, the wasps were individualized into new tubes with no water, no honey, and no hosts, and were held under controlled conditions as described in *Assay 1*. Mortality was recorded daily.

## Statistical analysis

Adult survival was estimated using the Kaplan-Meier survival analysis. Survival curves were compared using the log-rank (Mantel-Cox) test. After confirmation of normality (Shapiro-Wilk, D'Agostino & Pearson, and Kolmogorov-Sminorv's tests) (S2 Table) and homoscedasticity (Barllet's and Brown-Forsythe's) (S3 Table), Two-Way ANOVA followed by Tukey's test ($\alpha$ = 0.05) were applied to determine the effects of water and honey on wasp fecundity, emergence rate, sex ratio, and miscellaneous attacks. All estimates and analyses were performed using GraphPad Prism version 7.0b for Mac OS X (GraphPad Software, La Jolla, CA, USA).

## Supporting information

**S1 Table. Survival curve analysis (log-rank Mantel-Cox test) to evaluate the impact of constant and limited supply of water, honey, water + honey, and fasting (no honey, no water) on males, females, and host-provided females (females + hosts) of *Pachycrepoideus vindemmiae* (Hymenoptera: Pteromalidae).** Pupae of spotted-wing drosophila (*Drosophila suzukii*) (Diptera: Drosophilidae) were used as hosts. The numbers of replicates respectively for water, honey, water + honey, and fasting in the assay #1 were 10, 8, 9, and 8 for males; 13, 14, 16, and 10 for females; and 11, 14, 10, and 12 for females + hosts. In the assay #2 the number of replicates were 94 (water), 47 (honey), 95 (water + honey), 39 (no water, no honey), and 96 (host, SWD pupa). In both assays each replicate was formed by one mated individual wasp. (DOCX)

**S2 Table. Results of the normality tests (Shapiro-Wilk, D'Agostino & Pearson, and Kolmogorov-Sminorv's tests) for different treatments (water, honey, water + honey, and fasting) and life-history traits of the parasitoid *Pachycrepoideus vindemmiae* (Hymenoptera: Pteromalidae) reared on pupae of spotted-wing drosophila (*Drosophila suzukii*) (Diptera: Drosophilidae).** N = 11 (water), 14 (honey), 10 (water + honey), and 12 (fasting). Each replicate was formed by one mated individual wasp. (DOCX)

**S3 Table. Results of the homoscedasticity tests (Barllet's and Brown-Forsythe's tests) for different treatments (water, honey, water + honey, and fasting) and life-history traits of the parasitoid *Pachycrepoideus vindemmiae* (Hymenoptera: Pteromalidae) reared on pupae of spotted-wing drosophila (*Drosophila suzukii*) (Diptera: Drosophilidae).** N = 11 (water), 14 (honey), 10 (water + honey), and 12 (fasting). Each replicate was formed by one mated individual wasp.
(DOCX)

**S1 Video. *Pachycrepoideus vindemmiae* (Hymenoptera: Pteromalidae) attacking pupae of spotted-wing drosophila (SWD), *Drosophila suzukii* (Diptera: Drosophilidae), in laboratory.** Female wasp inserting her ovipositor through the SWD's pupal case and performing internal evaluation of the host prior oviposition. Note the wasp's ovipositor moving between the SWD pupal case and the pupa.
(MP4)

**S2 Video. *Pachycrepoideus vindemmiae* (Hymenoptera: Pteromalidae) attacking pupae of spotted-wing drosophila (SWD), *Drosophila suzukii* (Diptera: Drosophilidae), in laboratory.** Female wasp host-feeding on hemolymph of a SWD pupa following ovipositor withdrawal. Note the wasp's abdomen engorging as host-feeding takes place.
(MP4)

## Acknowledgments

We thank Dr. Gary Gibson (Canadian National Collection of Insects) for the wasp taxonomic identification; and Betsey Miller, Rachele Nieri, and Linda Brewer (Oregon State University) for the thorough review and insightful comments in the early versions of this manuscript.

## Author Contributions

**Conceptualization:** Cherre S. Bezerra Da Silva.

**Formal analysis:** Cherre S. Bezerra Da Silva.

**Funding acquisition:** Cherre S. Bezerra Da Silva, Vaughn M. Walton.

**Investigation:** Cherre S. Bezerra Da Silva, Briana E. Price, Alexander Soohoo-Hui, Vaughn M. Walton.

**Methodology:** Cherre S. Bezerra Da Silva, Briana E. Price, Vaughn M. Walton.

**Resources:** Vaughn M. Walton.

**Supervision:** Cherre S. Bezerra Da Silva, Vaughn M. Walton.

**Writing – original draft:** Cherre S. Bezerra Da Silva, Vaughn M. Walton.

**Writing – review & editing:** Cherre S. Bezerra Da Silva, Vaughn M. Walton.

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
