## [Decision Letter · Decision Letter 0]

21 Jun 2019

PONE-D-19-15049

Factors affecting the biology of Pachycrepoideus vindemmiae (Hymenoptera: Pteromalidae) on spotted-wing drosophila (Drosophila suzukii)

PLOS ONE

Dear Cherre Sade Bezerra Da Silva,

Thank you for submitting your manuscript to PLOS ONE. After careful consideration, we feel that it has merit but does not fully meet PLOS ONE’s publication criteria as it currently stands. Therefore, we invite you to submit a revised version of the manuscript that addresses the points raised during the review process.

We would appreciate receiving your revised manuscript by 13 August 2019. When you are ready to submit your revision, log on to https://www.editorialmanager.com/pone/ and select the ‘Submissions Needing Revision’ folder to locate your manuscript file.

To enhance the reproducibility of your results, we recommend that if applicable you deposit your laboratory protocols in protocols.io, where a protocol can be assigned its own identifier (DOI) such that it can be cited independently in the future. For instructions see: http://journals.plos.org/plosone/s/submission-guidelines#loc-laboratory-protocols

A rebuttal letter that responds to each point raised by the academic editor and reviewer(s). This letter should be uploaded as separate file and labeled ‘Response to Reviewers’.A marked-up copy of your manuscript that highlights changes made to the original version. This file should be uploaded as separate file and labeled ‘Revised Manuscript with Track Changes’.An unmarked version of your revised paper without tracked changes. This file should be uploaded as separate file and labeled ‘Manuscript’.

We look forward to receiving your revised manuscript.

Kind regards,

Kleber Del-Claro, PhD

Academic Editor

PLOS ONE

Journal Requirements:

Please ensure that your manuscript meets PLOS ONE’s style requirements, including those for file naming. The PLOS ONE style templates can be found at

Reviewers’ comments:

Reviewer’s Responses to Questions

**Comments to the Author**

1. Is the manuscript technically sound, and do the data support the conclusions?

Reviewer #1: Yes

Reviewer #2: Yes

2. Has the statistical analysis been performed appropriately and rigorously? 

Reviewer #1: No

Reviewer #2: Yes

3. Have the authors made all data underlying the findings in their manuscript fully available?

Reviewer #1: Yes

Reviewer #2: No

4. Is the manuscript presented in an intelligible fashion and written in standard English?

Reviewer #1: No

Reviewer #2: Yes

5. Review Comments to the Author

Reviewer #1: Review of the study entitled: “Factors affecting the biology of Pachycrepoideus vindemmiae (Hymenoptera: Pteromalidae) on spotted-wing drosophila (Drosophila suzukii)”

General comments

The goal of this manuscript was to evaluate the effects of “intrinsic” (sex and age) and “extrinsic” (resource offer; e.g. water, honey, water + honey) factors on the survival, fecundity, host-killing capacity, sex-ratio and emergence rate of a parasitoid wasp that lay eggs in fly pupae. In laboratory conditions, the authors have found that wasps fed with water + honey had increased rates of survival, that males and females had different survival rates depending on the offered food resource, that water resource is more important for the survival of young females, among other findings. The authors had performed two sets of experiments: for the first assay, they offered resources ad libitum, while for the second assay, the adult wasps were fed once during a 30-minutes period.

Although I find the study interesting and important due to its practical applications, I have struggled to understand the applied methods and results as I found it to be very confusing. In addition to it, the figures that I had access to have bad resolution and some of them (Figure 3) have illegible axes. I also had problems with some tables.

In Table 1, for instance, it is written: “Comparison of survival curves by long-rank (Mantel-Cox) test to evaluate the impact of constant and limited supply of water, honey, water + honey, and fasting (no honey, no water) on adult males and females of Pachycrepoideus vindemmiae (Hymenoptera: Pteromalidae), in the presence and absence of hosts…”, but when we look at the table, the results are separated by wasp’s sex and host presence, with no mention to any supply. Does that mean that the authors evaluated the effects of all supplies (water, honey, water + honey, and fasting) combined on males and females? By reading the corresponding result section, I don’t think so, since it is written: “…we found that the availability of water + honey resulted in significantly higher survival curves than honey alone, water alone, and fasting (no honey, no water), independent of sex and host availability (Males: P<0.0001, N=8-9; Females: P<0.0001, 129 N=10-16; Females + host: P=0.0049, N=9-15;) (Fig 2ab, Table 1). Here they discriminated the treatments, although Table 1 had not separated effects. It seems that the Female + host treatment (P=0.0049) considered the effects of water + honey in comparison to the other treatments (at least it is the same P value in the table). Furthermore, if they are comparing the water + honey treatment with the other three, it should not there be three P values for each sex treatment? (Female: water + honey x honey; water + honey x water; water + honey x fasting; Male: water + honey x honey…). The treatments got me confused, too. In the methods section, the authors established (for assay 1) eight treatments: (1) water, (2) honey, (3) water + honey, (4) no water, no honey (fasting), (5) water + hosts, (6) honey + hosts, (7) water + honey + hosts, and (8) no water, no honey + hosts. But in Table 1 the “hosts” treatments are not considered as supply, but in a treatment called Female + hosts. Although I believe that the authors did that since only females had the “host” treatment, it seems the treatments are changing throughout the text.

I also cannot understand the N used in the experiments. Most of the times the N shown are intervals. In Table 1 we have N=8-9; N=10-16 and N=9-15. What does an N of 9-15 mean? The reader must know exactly the N of each experiment. In Table 3, the legend reports an N = 8-16. However, the authors reported a DFd of 40 to 44, which means that the N was of at least 42 (DFd = N – a, where a = number of factors, which is 2, I believe). How are those values so different? Maybe there was some kind of pseudoreplication?

In Table 2, the host diet is “not applicable” to any treatments (male, female and female + hosts). I do not understand why, since the authors used hosts in half of the treatments described in assay 1, and there is even a treatment here that considered hosts. Is that because the authors did not offer only hosts in any treatment (assay 1)? If that is so, why is this treatment observed in assay 2? Also, why did the authors choose to conduct two assays? I saw no justification for that.

The abstract needs some improvements. The use of ‘host” as supply (together with water and honey) is confusing since I did not imagine that adults of P. vindemmiae feed on its hemolymph; I was thinking about the eggs inserted in the hosts. It should be clear that the adults feed on the pupae. This confusion happens during the text because sometimes it is not clear whether the authors are writing about the adult or juvenile wasps. I had to read the introduction and methods to understand the abstract. In addition, do the adults kill the pupae when they start to feed on it? Maybe this can be clarified in the introduction.

To conclude, I consider this manuscript to be promising, but the presentation of methods and results made me struggle to understand it. Although I see no major problems in the introduction and discussion, the manuscript still needs a lot of work and the ideas need to be more clear.

Specific comments

By reading the title, it is not possible to understand that the system involves a parasitoid species unless the reader knows the involved species. I had to read the abstract to understand it. I suggest the authors let this information explicit.

Line 33: Does “host-killing” mean the offspring kill? It would be good to define this term. I was confusing it with “host-feeding”.

Line 53: It needs a reference.

Lines 100-103: It seems a result part. If not, it needs a reference.

Lines 114-116: This sentence needs to be justified.

Line 159: Is the ± SD, SE?

Line 200: No need of the word parametric here. ANOVA is ways a parametric test.

Lines 204-205: The table needs to be formatted.

Lines 225-226: Should they be 0.4589 and 0.1835?

Lines 232-233: Is that the honey and water treatments alone or it includes water + honey treatment?

Line 253: If that is the standard error of the mean, it should be SE instead.

Line 257: Is that ± SE?

Line 269: 6-11 days cannot be on average.

Line 293: Why does it have intervals here (0.5-1; 17-19)?

Lines 346-347: The first half of the sentence has an odd construction.

Lines 354-357: I do not understand what do you mean here. Do you mean that there is no trade-off since the wasp does not kill the host?

Line 356: I saw both “trade-off” and “tradeoff” (line 375) throughout the manuscript. Please standardize.

Lines 383-386: it would be more informative if you explain right away that the water deprivation (not water availability) increased the host-killing.

Lines 388-391: Although the diet has not affected the emergence rates, it affected the number of viable eggs (Lines 327-330). Maybe these two parts can be gathered.

Lines 152, 155, 250, 539, 756: Please fix the term to “log-rank”.

Reviewer #2: The manuscript “Factors affecting the biology of Pachycrepoideus vindemiae (Hymenoptera: Pteromalidae) on spotted-wing drosophila (Drosophila suzukii)” describes a laboratory study that aimed to show factors that affect the biological aspects of P. vindemiae, as well as show the effectiveness of the included biological controller in situations with scarce nutritional resources. The relevance of water as a source and their influence on physiology and fecundity has been shown in a previous report by three authors of this paper “Water-Deprived Parasitic Wasps (Pachycrepoideus vindemmiae) Kill More Pupae of a Pest (Drosophila suzukii) as a Water-Intake Strategy”. Also the efficiency as biological controller over D suzukii has been tested by other studies (Bonneau et al., 2019). Therefore in those aspects I have not seen relevant information of the present manuscript. However, aspects related to sex and age related to longevity and parasite capacity throughout their lifespan are interesting aspects and new information produced by the study. In addition the parasitoid’s controlling capacity even in resource deficit, which is fundamental for the implementation of this parasitoid in integrated pest control systems and provides new opportunities for future research. The title of the paper I think is correct because it indicates that it will focus on the factors affecting the biology that has been adequately approached. And the objectives are well-defined, indicating which parameters they will address, but I missed a hypothesis indicating what they would expect given that they performed treatments with individuals supplying and omitting different resources, besides that there are already similar studies with other parasitoids.

- Line 175-177: “Fecundity was significantly increased by water deprivation (honey and fasting)…, while the effect of honey was insignificant”. I think that you missed to discuss these results (these are even mentioned in line 31-33 of the abstract), so I believe that it is necessary to discuss why water is increased fecundity in young females and honey was not shown significant differences, what could be the reason for that.

- Line 253: Table 2 is after table 3, which is in line 200, you need to place the tables in an ordered numbering.

- Line 467: “Emerged adults were released into new Bugdorm cages to restart the cycle”… Were individuals placed in new cages separately by sex? Or together? Please specify this.

- Line 486: “We tested 8-14 wasp/treatment/sex”. Could you describe it better. Were 8 to 14 wasps tested per treatment and per sex? Was that what you meant? That’s what I understood.

- Line 490-492: “To control for potential differences in the relative humidity among treatments that were water deprived (1, 3, 5, and 7) and water provided (2, 4, 6, and 8)”. The deprived water treatments were not 2,4,6,8 and those supplied with water were 1,3,5,7? that was confusing. On line 474 you mention the kinds of treatment and there you state that 1, 3, 5 and 7 were supplied with water and the other ones were deprived of water.

- Line 515 - 520: In my view, I recommend that for a better understanding this paragraph should be at the beginning, in line 472, because if not, it sounds like you’re coming back on a subject that you’ve already explained and with no order in the text.

- Supporting information - S1_table: I consider that means, medians and variances of the data are not included, as well as the normality and homocedasticity tests of the analyzed data.

6. PLOS authors have the option to publish the peer review history of their article (what does this mean?). If published, this will include your full peer review and any attached files.

Reviewer #1: No

Reviewer #2: No

---

## [Author Response · Author response to Decision Letter 0]

30 Jun 2019

Most of the suggestions/comments by the two reviewers have been implemented by us the authors, improving significantly our manuscript. Many thanks!

---

## [Decision Letter · Decision Letter 1]

10 Jul 2019

Factors affecting the biology of Pachycrepoideus vindemmiae (Hymenoptera: Pteromalidae), a parasitoid of spotted-wing drosophila (Drosophila suzukii)

PONE-D-19-15049R1

Dear Dr. Cherre Sade Bezerra Da Silva,

We are pleased to inform you that your manuscript has been judged scientifically suitable for publication and will be formally accepted for publication once it complies with all outstanding technical requirements.

Shortly after the formal acceptance letter is sent, an invoice for payment will follow. To ensure an efficient production and billing process, please log into Editorial Manager at https://www.editorialmanager.com/pone/, click the “Update My Information” link at the top of the page, and update your user information. If you have any billing related questions, please contact our Author Billing department directly at authorbilling@plos.org.

With kind regards,

Kleber Del-Claro, PhD

Academic Editor

PLOS ONE

Reviewers’ comments:

Reviewer’s Responses to Questions

**Comments to the Author**

1. If the authors have adequately addressed your comments raised in a previous round of review and you feel that this manuscript is now acceptable for publication, you may indicate that here to bypass the “Comments to the Author” section, enter your conflict of interest statement in the “Confidential to Editor” section, and submit your “Accept” recommendation.

Reviewer #1: All comments have been addressed

Reviewer #2: All comments have been addressed

2. Is the manuscript technically sound, and do the data support the conclusions?

Reviewer #1: Yes

Reviewer #2: Yes

3. Has the statistical analysis been performed appropriately and rigorously? 

Reviewer #1: Yes

Reviewer #2: Yes

4. Have the authors made all data underlying the findings in their manuscript fully available?

Reviewer #1: Yes

Reviewer #2: Yes

5. Is the manuscript presented in an intelligible fashion and written in standard English?

Reviewer #1: Yes

Reviewer #2: Yes

6. Review Comments to the Author

Reviewer #1: The authors have performed a great effort to address all issues and I believe the manuscript is now much more clear. I just want to specify one of my comments since it was not clear enough. In table 3, the header is not centered like the other tables (at least they seem centered to me). Also, in the fourth column (table 3), the values are sometimes spaced and sometimes not. Other than that, I have nothing more to add.

Reviewer #2: (No Response)

7. PLOS authors have the option to publish the peer review history of their article (what does this mean?). If published, this will include your full peer review and any attached files.

Reviewer #1: No

Reviewer #2: No

---

## [Editor Report · Acceptance letter]

15 Jul 2019

PONE-D-19-15049R1

Factors affecting the biology of *Pachycrepoideus vindemmiae* (Hymenoptera: Pteromalidae), a parasitoid of spotted-wing drosophila (*Drosophila suzukii*)

Dear Dr. Bezerra Da Silva:

I am pleased to inform you that your manuscript has been deemed suitable for publication in PLOS ONE. Congratulations! Your manuscript is now with our production department.

With kind regards,

on behalf of

Dr. Kleber Del-Claro

Academic Editor

PLOS ONE